# Molecular Transmission Network of Newly Reported HIV Infections in Pengzhou, Sichuan Province: A Study Based on Genomics and Spatial Epidemiology

**DOI:** 10.3390/ijerph20032523

**Published:** 2023-01-31

**Authors:** Dan Yuan, Xia Zhong, Yiping Li, Qinying He, Na Li, Hanqi Li, Yang Liu, Ling Li, Linglin Zhang, Yi Yang, Shu Liang

**Affiliations:** 1Center for AIDS/STD Control and Prevention, Sichuan Center for Disease Control and Prevention, Chengdu 610044, China; 2School of Management, Chengdu University of Traditional Chinese Medicine/Healthy Sichuan Research Institute, Chengdu 611137, China; 3Center for AIDS/STD Control and Prevention, Chengdu Center for Disease Control and Prevention, Chengdu 610047, China; 4Center for AIDS/STD Control and Prevention, Pengzhou Center for Disease Control and Prevention, Chengdu 611930, China; 5Department of Geography & Anthropology, Louisiana State University, Baton Rouge, LA 70803, USA

**Keywords:** HIV-1, molecular network, spatial analysis

## Abstract

Objective: The objective of this study was to understand the molecular transmission characteristics of newly reported HIV infections in the city of Pengzhou, Sichuan Province, to analyze the risk factors of transmission network and spatial clustering and the transmission characteristics, and to provide a scientific basis for precision prevention and intervention. Methods: Anticoagulated whole blood was collected from newly reported HIV infections in Pengzhou from March 2019 to August 2021. After the plasma was isolated, the HIV-1 pol gene was amplified and sequenced by reverse transcriptase polymerase chain reaction (PCR). The obtained gene sequences were used to construct a maximum likelihood phylogenetic tree for the analysis of virus subtypes, and a molecular transmission network was constructed using the genetic distance method to evaluate the transmission pattern of people living with HIV/AIDS in Pengzhou. A logistic regression model was used to find out the potential risk factors for entering the molecular transmission network with the number of nodes ≥ 2. Spatial analysis is used to show the geographical pattern of the proportion of newly reported HIV infections entering the molecular transmission network, and a flow map is used to show the intensity of transmission within and between townships. Results: A total of 463 newly reported HIV-infection sequences were obtained in this study, including 237 cases (51.19%) of CRF01_ AE, 159 cases (34.34%) of CRF07_BC, 45 cases (9.72%) of B, 15 cases (3.24%) of CRF08_BC and 7 cases (1.5%) of others. The number of clusters was the highest when the gene distance was 0.009, with a total of 246 sequences entering the network, forming 54 clusters, and the network entry rate was 55.36%. There were 170 sequences with more than two nodes in the network sequence. The logistic regression showed that compared with age < 50 years old, age ≥ 50 years old has a higher risk of transmission (OR = 3.43, 95% CI = 2.06–5.71); compared with farmers, the risk of transmission within industry is lower (OR = 0.046, 95% CI = 0.25–0.87); and compared with CRF07_BC, CRF01_AE (OR = 6.09, 95% CI = 3.60–10.30) and B (OR = 20.31, 95% CI = 8.94–46.13) had a higher risk of transmission. Men aged ≥ 50 years are mainly clustered with women between 50 and 70 years of age. In addition to being clustered with gay men, there are nine (50%) and three (16.7%) chains of transmission between gay men and heterosexual men and women, respectively. In the geographical space, there is no hot spot clustering of the molecular propagation network. The subtype B was mainly distributed in the town of Tianpeng and formed transmission networks in eastern Pengzhou;0020CRF01_AE is mainly distributed in the town of Lichun and formed transmission networks in the west and north of Pengzhou. Conclusion: This study reveals the characteristics and influencing factors of molecular network transmission in the region, as well as the spatial transmission characteristics of newly reported HIV infections in recent years, and reveals the geographical differences in HIV-1 transmission. The results provide a scientific basis for the development of local AIDS-specific intervention measures.

## 1. Introduction

Acquired Immune Deficiency Syndrome (AIDS) is an important infectious disease that endangers human health. It has caused nearly 84.2 million infections in people worldwide. According to the The Joint United Nations Programme on HIV/AIDS (UNAIDS) there were about 38.4 million people living with HIV/AIDS (PLWH) worldwide in December 2021, of which 10 million were not receiving antiviral treatment and 1.5 million were newly reported HIV infections [1,2]. By the end of December 2021, there were 1.147 million PLWH cases in China (excluding foreign countries, Hong Kong, Macao and Taiwan Special Administrative Regions, the same below), of which 1.063 million received standardized antiviral treatments (92.68%), 74.4% were male and sexual transmission accounted for 90.75% [3,4]. As a serious epidemic area, Sichuan province reported 210,000 PLWH cases by December 2021, including 18,000 newly reported HIV infections in 2021. Of the newly reported HIV infections, 3735, which is 20.57% of the whole Sichuan province, came from Chengdu, the capital city of Sichuan province. Pengzhou known as “Little Chengdu” is a county level city with a long history in Chengdu and is one of the birthplaces of ancient Shu culture. It is also a city integrating business, culture, consumption and tourism. A total of 720 HIV infections were reported in Pengzhou from March 2019 to August 2021. Previous studies have shown that the spatial distribution of the epidemic is gradually shifting from the central urban area to the northern suburbs, among which Pengzhou is the hot spot of the epidemic [5]. Therefore, taking Pengzhou as the pilot area of this study can be representative of the HIV epidemics in Chengdu.

The molecular network uses the genetic similarity between viral gene sequences to infer viral transmissions, starting from the previous social or sexual transmission network construction relying solely on questionnaires and peer tracking, to the later molecular transmission network established by using the genetic information of HIV sequences [6,7], which has been applied to the study of HIV transmission networks [8]. Zhang et al. used molecular networks to quantify the local HIV transmission pattern in Hangzhou, providing onsite evidence for a precise HIV prevention strategy [9]. Lebedev et al. used molecular networks to analyze the HIV-1 transmission clusters in the Republic of Uzbekistan to determine the transmission networks and the transmission clusters [10]. Using molecular networks to construct the macroscopic transmission network of PLWH is more accurate and has been widely used to verify the results and conclusions of epidemiological field investigations. Spatial geographic variation is an important factor affecting the pattern of HIV transmission [11]. The use of spatial epidemiological analysis helps to understand the dynamic changes of HIV-1 subtypes from a spatial perspective and is currently widely used to infer the transmission relationship across time, space and population. Jiang et al. analyzed the molecular network and spatial geography to study the hotspots and cross-regional transmission characteristics of HIV transmission in the Guangxi Zhuang autonomous region, and provided the basis for formulating a precise HIV prevention strategy in the region [10]. Nduva et al. combined molecular network and geographic information to quantify the HIV-1 flow rate between key populations (men having sex with men (MSM), people who inject drugs and female sex workers) and general heterosexual populations and between different geographic regions [12]. In this study, a molecular transmission network analysis combined with a spatial epidemiological analysis was used to explore the spatial distribution pattern of the HIV/AIDS epidemic in the Pengzhou area, identify hot spots and reveal the spatial distribution characteristics and epidemic rules of the HIV/AIDS epidemic. In addition, a regression model was used to analyze the influencing factors of HIV transmission in this area, so as to provide a basis for formulating scientific and regional AIDS prevention and control policies. At the same time, through the analysis results from the Pengzhou area, this study can explore whether this method is applicable to precise public health interventions in other areas.

## 2. Materials and Methods

### 2.1. Research Object

The Sichuan registered population is ranked third in the country; the resident population ranks fourth in the country. Since 2010, the number of permanent residents in Sichuan has been increasing year by year, reaching 83.71 million in 2020, accounting for 5.93% of the total population of the country, and population urbanization accounts for 56.73%. The population of Chengdu was expected to reach 20,947,000 in 2020, accounting for 25.02% of the total population of the province, and population urbanization accounts for 78.77% [13]. Pengzhou’s resident population reached 770.4 thousand in 2020, accounting for 3.73% of Chengdu’s total population, and population urbanization accounted for 49.13%.

From March 2019 to August 2021, there were a total of 720 newly diagnosed PLWH cases, of which 484 were recruited, accounting for 67.22%. The information on newly reported HIV infections was screened from the Sichuan Provincial Center for Disease Control (CDC) and Prevention AIDS Prevention and Control Basic Information System. For these newly reported HIV infections, the patients’ information was derived from medical records in the system, including social demographics (including age, gender, ethnicity, marital status), main routes of HIV transmission (heterosexual, homosexual), laboratory-related test data (pre-treatment CD4 count, virus subtype, drug resistance).

The reference strains for this study are from the Los Alamos HIV sequence database [https://www.hiv.lanl.gov/ (accessed on 23 November 2022)]. Other datasets used and/or analyzed in the current study are available upon reasonable request from the respective authors.

All participants signed a written informed consent form prior to the sample donation requesting strict adherence to the principles of data confidentiality, together with reasonable access to the research results, and all researchers signed data confidentiality agreements.

### 2.2. Sample Collection and Preservation

For each newly reported HIV infection, 5 mL of venous whole blood was collected using 8 mL EDTA-K2-containing vacuum blood collection tubes for a CD4+ T cell count and HIV-1 genotype resistance testing. The Pengzhou CDC conducted CD4+ T cell count detection on the collected anticoagulant whole blood. The isolated plasma was sent to the Sichuan Provincial Center for Disease Control and Prevention for HIV-1 genotype resistance testing under frozen conditions, and the samples were stored at −80 °C in a refrigerator before testing.

### 2.3. HIV-1 Genotype Drug Resistance Detection

Roche’s automatic virus nucleic acid extractor (MagNA Pure96) and Roche’s MagNA Pure 96 DNA and Viral NA Small Volume Kit were used to extract the viral nucleic acid, in strict accordance with the instrument instructions. A nested reverse transcription fluorescence quantitative polymerase chain reaction and nested PCR were used to amplify the 1300 bp of the HIV-1 pol gene region, including the first 300 amino acid sequences of protease and reverse transcriptase. Two rounds of primers and PCR amplification conditions were used, in accordance with the HIV-1 Genotype Drug Resistance Detection and Quality Assurance Guidelines (2013 Edition) [14].

### 2.4. Sequence Arrangement and Subtype Classification

The sequence analysis software Sequencer 4.9 was used to edit, clean and splice the sequence. The sequence alignment results were corrected with the software Bioedit and compared with the Los Alamos HIV Database reference sequence. Then the Neighbor-Joining method Kimura2-parameter model of the Mega6.0 software was used to construct the phylogenetic tree, which was repeatedly constructed 1000 times. A bootstrap value > 70% was used to verify the accuracy of genotyping and to determine genotyping.

### 2.5. The Construction of a Molecular Network Transmission Model

Using HyPhy 2.2.4 software to calculate the gene distance between genes under the TN93 model, according to the HIV transmission network monitoring and intervention technical guidelines (2021 trial version) [15], within the gene distance of 1.5%, the gene distance value with the largest number of clusters in the network sequence was used as the gene distance threshold of the network. Finally the visualization of the transmission network was realized using Cytoscape 3.6.1 software [5], forming a number of potential molecular clusters. The network characteristics were determined, including nodes (individuals in the network), edges (links between two nodes, representing the potential transmission relationship between two individuals), degrees (the number of edges connecting one node to other nodes), network size (the number of individuals in the cluster) and clusters (groups of link sequences).

### 2.6. Statistical Analysis

SPSS 19.0 was used for the statistical analysis. Univariate and multivariate logistic regression models were used to screen the factors with two or more nodes in the molecular network. A univariate logistic regression analysis was performed on each variable, and variables with *p* < 0.2 were included in the multivariate logistic regression analysis. *p* < 0.05 was considered statistically significant.

### 2.7. Moran’s I Index of Local Spatial Autocorrelation

In the clustering graph, the color represents the network access rate, and the circle size represents the number of networks. The larger the circle, the greater the number of networks; the border color represents the spatial clustering pattern of the access rate. “High-high” means that a given area with a high access rate is surrounded by areas with a high access rate; “low-low” refers to an area with a low access rate surrounded by an area with a low access rate; “low-high ” refers to an area with a low network access rate surrounded by an area with a higher network access rate; “high-low” refers to an area with a high access rate surrounded by an area with a relatively low access rate; a value of zero means that the access rate is randomly distributed between regions (that is, there is no spatial clustering pattern). A z-test was performed to determine whether each spatial clustering pattern of the access rate was significantly different from a random distribution. All the analyses were performed in ArcGIS (version 10.3) [16].

Individuals linked in the molecular network transmission cluster were used to reflect whether there was more HIV transmission within or between regions in the region. In order to show more detailed HIV transmission between cities, the transmission network of each HIV-1 subtype was visualized. The links between regions were represented by lines of different colors. A darker color and a thicker line indicate a stronger the relationship between HIV infections in these two regions; the links within a region are represented by blue circles. A darker color and a thicker circle indicate a stronger internal relationship of HIV infections in this region; the size of the red circle indicates the number of infected people in the region. The larger the circle, the more infected people. The transport diagram was created in QGIS [13] (version 3.10) [17].

## 3. Results

### 3.1. Basic Situation

A total of 484 participants with newly reported HIV infections were recruited and 463 sequences were successfully obtained, including for 348 males (75.16%), 342 individuals aged ≥50 (73.87%), 368 farmers (79.48%), 218 individuals of primary school education level (47.08%), 291 wedlock individuals (62.85%), 443 cases of heterosexual transmission (95.68%), 443 cases with no drug resistance (95.68%), and 243 cases with a first CD4+ cell count > 200/mm^3^ (52.48%) (Table 1).

A total of eight subtypes were detected, including CRF01_AE (237 cases (51.19%)), CRF07_ BC (159 cases (34.34%)), B (45 cases (9.72%)), CRF08_ BC (15 cases (3.24%)) and CRF55_01B (4 cases (0.86%)). In addition, one case each of B/CRF01_AE, C and CRF60 _BC (0.22%) were detected, see Figure 1. It was found that when the genetic distance was 0.009, the total number of transmission clusters in the network reached a peak, and 252 sequences entered the network, with a network entry rate of 54.43%, see Figure 2. A total of 55 transmission clusters were found, and the number of sequences in each cluster ranged from 2 to 34. Among them, four larger clusters had 34, 25, 21 and 20 nodes, respectively, accounting for 39.68% of the total number of clustered cases, as shown in Figure 3. The CRF01_AE subtype had the largest number of clusters (34), including 3 larger clusters. The subtype B formed two clusters, one of which had 34 nodes. The CRF07_BC subtype also had many clusters, and the largest cluster had six nodes. The CRF08_BC subtype formed only one cluster.

### 3.2. Influencing Factors of HIV-1 Molecular Transmission Network

Among the cases analyzed, 170 had a node degree of 2 or more, mainly those who were male (81.18%), aged ≥ 50 years old (87.06%), farmers (87.65%), of primary school education level (54.71%), wedlock (61.18%) and cases of heterosexual transmission (97.65%), the CRF01_AE strain (67.06%), with no drug resistance (98.24%) and a first CD4+ cell count > 200/mm^3^ (52.35%). A logistic regression model was used to analyze the influencing factor of a node degree ≥ 2 on the network. The results showed that the effect of age ≥ 50 years old was 3.43 times that of age < 50 years old (95% CI = 2.06–5.71), employed status was 0.046 times that of farmers (95% CI = 0.25–0.87), CRF01_ AE and B were 6.09 times (95% CI = 3.60–10.30) and 20.31 times (95% CI = 8.94–46.13) that of CRF07_BC, respectively. See Table 1 for details.

### 3.3. Characteristic Analysis of HIV Molecular Transmission Network

Men aged < 50 were more connected to women aged 60–70 (13/19), men aged 50–70 and 80 were more connected to women aged 50–70 (138/146), and men aged 70–80 were more connected to women over 80 (6/45), in addition to women aged 50–70 (37/45) (Figure 4a).

In the molecular cluster of newly reported HIV infections among MSM, there are six edges connected between newly reported HIV infections in MSM, nine edges connected between newly reported HIV infections in MSM and newly reported HIV infections in heterosexual males, and three edges connected between newly reported HIV infections in MSM and newly reported HIV infections in heterosexual females (Figure 4b).

### 3.4. Spatial Analysis of HIV Molecular Transmission Network

For the spatial analysis, four gene subtypes (CRF01_AE, B, CRF07_BC and CRF08_BC) were distributed in all 20 townships of Pengzhou city.

The number and geographical distribution of the different subtypes of newly reported HIV infections vary. CRF01_AE was the most widely distributed (in 19 towns except for the town of Bailu) and was most widely distributed in the towns of Lichun (40/162) and Tianpeng (20/162). Although the number of newly reported subtype B HIV infections was not large (45 cases), the network entry rate was high (80%), mainly concentrated in the town of Tianpeng (13/14). The number of CRF07_BC subtypes was large (159 cases), but the network entry rate was low (30.8%), mainly concentrated in the town of Mengyang. Only five cases (33.3%) of the subtype CRF08_BC entered the network, distributed among five towns.

The number of network access channels, the network access rate and cluster numbers for newly reported HIV infections in different towns were also different. There were 15 molecular clusters related to the town of Tianpeng (27.28%), followed by 13 molecular clusters related to the town of Mengyang (24.07%). The access rate for newly reported HIV infections in the town of Lichun was the highest (75.93%). In addition, the towns of Longmenshan, Xinxing and Xiaoyudong, which had a lower number of networks, also had higher access rates of 75%, 66.67% and 66.67%, respectively (Appendix A). From the clustering results for the network access rate, the town of Gexianshan was shown to have a significant low-high access rate and Mengyang had a low-low access rate (Figure 5a).

By mapping the connections between individuals with newly reported HIV infections and the towns in which they lived, we found that most individuals with newly reported HIV infections had intra-township connections, and the strength of the connections within and between towns was usually different (Figure 5b). Overall, the most significant connections were in Tianpeng town; its internal association and cross-town association were strong, with 91 edges and 330 edges, respectively. In terms of cross-town associations, all towns had a cross-town-association ratio of more than 60%, except Bailu. In terms of subtype analysis, the town-level association of the subtype CRF01_AE was mainly concentrated in the southwest of Pengzhou, and 54 connections were found within the town of Lichun, while the cross-town-level association was mainly between Lichun and Tianpeng (Figure 5c). The town association of the subtype B was mainly concentrated in the east-central area of Pengzhou, while 75 connections were found within Tianpeng Town, and 228 connections were directed to the surrounding towns (Figure 5d). There were fewer newly reported HIV infections across towns in terms of the subtype CRF07_BC, and six connections were found in the easternmost town of Mengyang (Figure 5e). Due to the small number of subtype CRF08_BC entering the network, there were only one connection between the towns of Guihua and Zhihe, and two connections between the towns of Aoping, Sanjie and Mengyang (Figure 5f). The connections within and between townships are shown in Table 2.

## 4. Discussion

In this study, we found that CRF01_AE (51.19%) and CRF07_BC (34.34%) were the main epidemic subtypes in Pengzhou, which was consistent with the previous studies carried out in China and Sichuan province [18,19,20,21]. We also found a variety of low epidemic strains: B (9.72%), CRF08_BC (3.24%), CRF55_01B (0.86%), B/CRF01_AE (0.22%), C (0.22%) and CRF60_BC (0.22%). As for the molecular network entry rate, the highest in this study was for subtype B, which was different from China and Sichuan province as a whole, where the subtypes CRF07BC and CRF01AE had the highest molecular network entry rate [22,23,24]. This suggests a diversified source of local HIV-1 infection, possibly due to the mobility of newly reported HIV infections, relatively developed transportation and more frequent economic activities. The epidemic situation brings big challenges for local HIV prevention and control. The real-time monitoring of high-risk groups using a molecular transmission network should be strengthened, not only for dominant epidemic subtypes, such as CRF01_AE and CRF07_BC, but also for other subtypes, such as the nationally low epidemic level but locally aggregated subtype B. Moreover, by combining the macro-monitoring of traditional epidemiology with the micro-analysis method of the molecular transmission network, it is most effective to find out the emergence and prevalence of different HIV-1 subtype viruses, and to achieve precise prevention and control.

The results of the multivariate analysis showed that the risk of HIV transmission for the subtypes CRF01_AE and B was higher than for CRF07_BC among individuals with PLWH aged ≥ 50 years old. The risk of HIV transmission in the employed is low. Employed individuals with newly reported HIV infections have more access to AIDS knowledge than farmers, so the risk of HIV transmission is relatively low. It has been reported [25] that the number and composition ratio of reported cases of new HIV infections in individuals aged ≥ 50 years old in Pengzhou are increasing year by year. In recent years, the elderly have been the fastest growing group of HIV infected individuals in China [26] and in Sichuan [27,28]. Due to ae lack of self-protection awareness among the elderly, the rate of condom use and the level of AIDS-related knowledge are low, which can easily lead to high-risk sexual behavior, so the risk of infection and transmission in the elderly population is further increased [29,30]. Targeted health education and prevention interventions in line with the behavioral and psychological characteristics of the elderly need to be developed to support policies to reduce the spread of HIV in the population, and ultimately to promote health. The risk of transmission of newly reported HIV infections infected of the subtype CRF01_AE in this area is higher than that of newly reported HIV infections of the subtype CRF07_BC, while the CD4(+) T-cell recovery of newly reported HIV infections of the subtype CRF01_AE is worse than that of subtype CRF07_BC infections. The main reason for this is the higher proportion of X4 tropic viruses [30,31], which will bring great challenges for the follow-up treatment of newly reported HIV infections of the subtype CRF01_AE. Therefore, the monitoring of subtype CRF01_AE transmission in this area should be strengthened to reduce the prevalence of this subtype in this area. It is worth noting that subtype B, the first prevalent subtype, was no longer the main epidemic subtype in our province [32,33], but in this study 80% of the subtype B entered the transmission network and the correlation degree of the entry network was above 9, indicating that subtype B in this area is more active and clustered. It is necessary to strengthen the investigation of traceability and the long-term dynamic monitoring of newly reported HIV infections of the subtype B to curb the further spread of the virus as soon as possible.

The study found that in the molecular transmission network, men in the ≥ 50 years old group were mainly connected to women in the same age group, not to women in younger age groups, indicating that men and women aged ≥ 50 years old are still sexually active and have a high risk of HIV transmission [33]. In addition, we also found that in the molecular transmission network, in addition to being connected to other MSM, MSM were also connected to heterosexual male and female newly reported HIV infections. Possibly during an epidemiological investigation, MSM individuals with newly reported HIV infections do not report their real route of infection due to the associated stigma and a discriminatory environment, and the fact that some of them also have sex with women. A report in the United States found that 4% of MSM in the US National Surveillance System were associated with women, but through the molecular transmission network survey, it was found that these MSM were associated with 29% of women infected for the first time [34].

Among all newly reported HIV infections, the high rate of network access was mainly concentrated in the central and northern regions of Pengzhou, especially in the towns of Lichun and Aoping. The rate of network access and the number of clusters among newly reported HIV infections were higher, indicating that the transmission of HIV in this area is more active. It is necessary to strengthen HIV/AIDS interventions in the two places. The town of Tianpeng, which has more clusters, has a relatively low rate of network access. It may be that the region is a central town with a relatively developed economy and a more floating population [35]. The main members of the transmission network are from other neighboring areas. The geographical clustering analysis of the network access rate showed no high-high clustering, indicating that there was no obvious geographical aggregation of HIV transmission in the region in recent years.

The intensity of transmission inter-township and intra-township reflects the geographic transmission characteristics of the main HIV subtypes prevalent in the area. This study found that there are several characteristics of HIV transmission between towns, which are worth discussing in order to better understand local HIV infection. First, attention should be paid to towns with strong regional transmission (such as between Lichun and Tianpeng). The transmission links between villages and towns that are geographically far apart are even closer than those between villages and towns that are close to each other, which means that local health departments in these villages and towns should increase their information exchange in order to better carry out joint interventions (such as between Tianpeng and Tongji and Lichun and Longmenshan). The possible causes of HIV transmission between regions are usually determined by transmission between sexual partners from different residential communities or by the migration of newly reported HIV infections [28], and reflect the high frequency of sexual activity among older populations in the region. Secondly, targeted interventions should be taken according to the population characteristics of transmission between townships. This study found that the proportion of transmission between most villages and towns is high, reaching more than 85%. The proportion of transmission between villages and towns including Lichun, Mengyang and Tianpeng is relatively low, but mainly based on transmission between villages and towns, which indicates that HIV prevention in areas of population and economic concentration requires joint internal and external intervention (cooperation with other regions). Thirdly, attention should be paid to the epidemic dynamics of HIV-1 subtypes. The transmission map in this study shows that some townships are key sites for virus transmission and frontline areas in need of urgent intervention (such as for the CRF01_AE subtype and CRF07_B subtype).

### Limitations

This study only included local newly reported HIV infections for analysis, and lacked previous case data. Secondly, the analysis data is only from the city of Pengzhou, so there is a lack of correlation analysis with other regions and a lack of preventive intervention resource data. The third is that there is a lack of sufficient research into the standards of variable division in the Pengzhou area, which may cause a certain degree of bias. This situation may result in an incomplete molecular transmission network. The research team will continue to carry out molecular network monitoring of the population while expanding the molecular monitoring population, to discover how new characteristics of the epidemic change over time and provide a basis for adjusting the prevention and control measures of the AIDS epidemic.

## 5. Conclusions

Our study re-emphasized the use of spatial epidemiological methods to describe the HIV-1 molecular transmission network [9], reflecting the spread of HIV between villages and towns in pengzhou, Sichuan. This study identified the high-risk groups in the region, especially the HIV transmission of PLWH in individuals aged ≥ 50 years old and identified the main epidemic virus subtypes in the region in recent years as CRF01_AE and B, with subtype B being highly clustered, forming a local short-term outbreak trend. It reveals the trends of local HIV transmission within and between townships. According to the results of the study, the method of multidisciplinary linkage can find more local epidemic characteristics, providing a basis for health administrative departments to strengthen precise prevention and control interventions.

## Figures and Tables

**Figure 1 ijerph-20-02523-f001:**
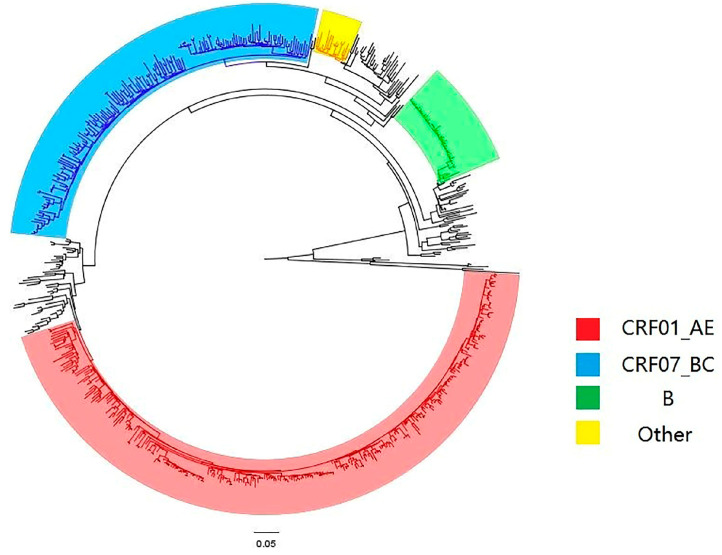
Phylogenetic tree diagram of the system.

**Figure 2 ijerph-20-02523-f002:**
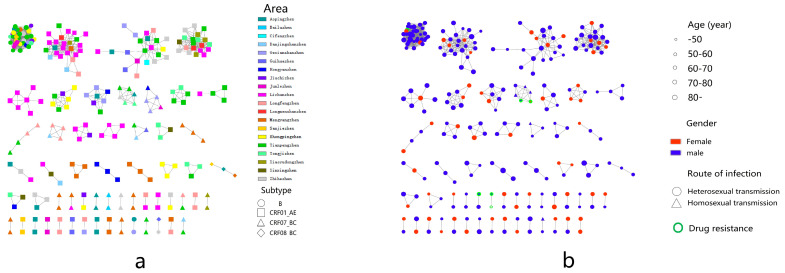
Molecular network diagram. (**a**) Township and subtype virus molecular transmission network. (**b**) Age, sex, transmission route and drug resistance in the molecular transmission network.

**Figure 3 ijerph-20-02523-f003:**
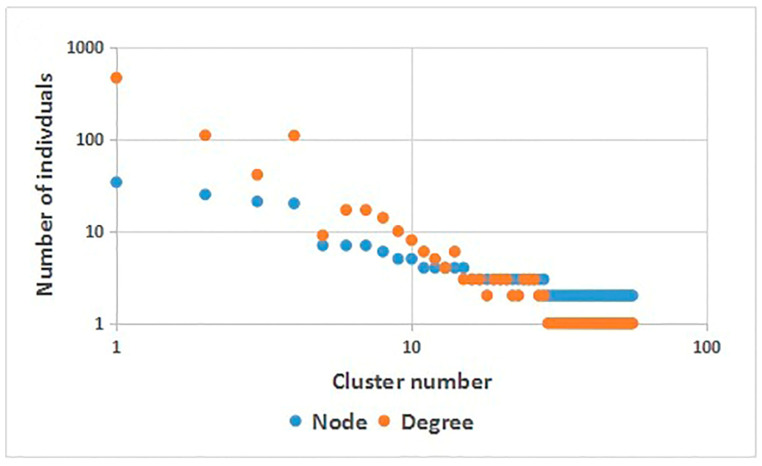
Molecular network parameters.

**Figure 4 ijerph-20-02523-f004:**
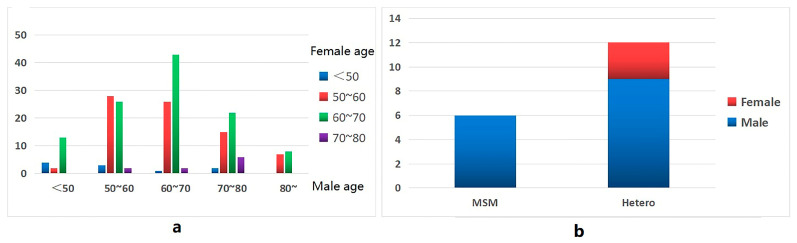
Propagation characteristic diagram of the molecular network. (**a**) Male and female connections at different ages. (**b**) Connections between MSM and patients with heterosexual transmission and patients of other genders.

**Figure 5 ijerph-20-02523-f005:**
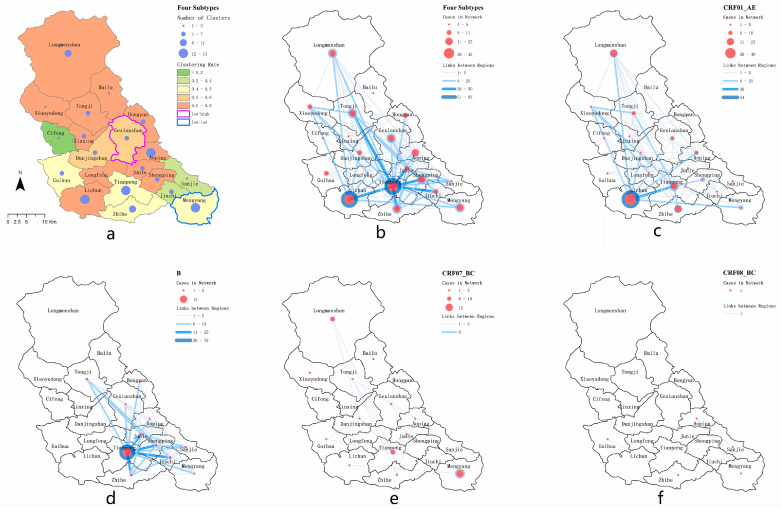
Spatial clustering and spatial transmission graph of the molecular propagation network. (**a**) Township area and cluster analysis, (**b**) spatial transmission of strain subtypes, (**c**) spatial transmission of strain subtype CRF01_AE, (**d**) spatial transmission of strain subtype B, (**e**) spatial transmission of strain subtype CRF07_BC, (**f**) spatial transmission of strain subtype CRF08_BC.

**Table 1 ijerph-20-02523-t001:** Analysis of the influencing factors of the Pengzhou molecular network cluster nodes ≥ 2.

Classification	Node ≥ 2		Total	n%	OR (95%CI) ^a^	AOR (95%CI) ^b^
	n	%				
**Gender**						
male	138	81.18	348	75.16	1	1
female	32	18.82	115	24.84	0.59 (0.37~0.93) **	0.66 (0.39~1.12) *
**Age**						
<50	22	12.94	121	26.13	1	1
≥50	148	87.06	342	73.87	3.43 (2.06~5.71) ***	2.10 (1.13~3.92) **
**Occupation**						
farmer	149	87.65	368	79.48	1	1
employed	21	12.35	95	20.52	0.42 (0.25~0.71) **	0.46 (0.25~0.87) **
**Education Level**						
illiteracy	24	14.12	47	10.15	1	1
primary school	93	54.71	218	47.08	0.71 (0.38~1.34)	0.82 (0.39~1.69)
junior high school and above	53	31.18	198	42.76	0.35 (0.18~0.67) **	0.61 (0.28~1.34) *
**Marital status**						
single/divorced/widow	66	38.82	172	37.15	1	
wedlock	104	61.18	291	62.85	0.89 (0.61~1.32)	
**Route of infection**						
heterosexual transmission	166	97.65	443	95.68	1	1
homosexual transmission	4	2.35	20	4.32	0.42 (0.14~1.27) *	1.64 (0.45~5.95) *
**Subtype**						
CRF07_BC	21	12.35	159	34.34	1	1
CRF01_AE	114	67.06	237	51.19	6.09 (3.60~10.30) ***	5.25 (3.04~9.07) ***
B	34	20	45	9.72	20.31 (8.94~46.13) ***	19.08 (8.03~45.32) ***
CRF08_BC	1	0.59	15	3.24	0.47 (0.06~3.76)	0.39 (0.05~3.20) *
other	0	0	7	1.51		
**Drug resistance**						
YES	3	1.76	20	4.32	1	1
NO	167	98.24	443	95.68	3.43 (0.99~11.88) *	3.85 (0.92~16.03) *
**First CD4+ T cell count**						
≤200	73	42.94	203	43.84	1	
>200	89	52.35	243	52.48	1.03 (0.70~1.52)	
unknown	8	4.71	17	3.67		

* *p* value < 0.2; ** *p* value < 0.05; *** *p* value < 0.001. ^a^: single factor analysis, ^b^: multivariate analysis.

**Table 2 ijerph-20-02523-t002:** Analysis of the results of inter-township and intra-township connections of newly reported HIV infections among towns in Pengzhou.

Town	All			CRF01_AE			CRF07_BC			B			CRF08_BC		
	Intra-Township	Inter-Township	Proportion of Transmission Inter-Township (%)	Intra-Township	Inter-Township	Proportion of Transmission Inter-Township (%)	Intra-Township	Inter-Township	Proportion of Transmission Inter-Township (%)	Intra-Township	Inter-Township	Proportion of Transmission Inter-Township (%)	Intra-Township	Inter-Township	Proportion of Transmission Inter-Township (%)
Cifengzhen	0	14	100	0	2	100	0	0	/	0	0	/	0	0	/
Danjingshanzhen	0	30	100	0	12	100	0	6	100	0	0	/	0	0	/
Xinxingzhen	0	14	100	0	14	100	0	0	/	0	0	/	0	0	/
Jiuchizhen	2	92	97.87	0	16	100	0	0	/	2	76	97.44	0	0	/
Longmenshanzhen	1	38	97.44	1	38	97.44	0	0	/	0	0	/	0	0	/
Sanjiezhen	2	54	96.43	1	0	0	0	0	/	1	53	98.15	0	1	100
Junlezhen	4	93	95.88	1	5	83.33	0	3	100	3	85	96.59	0	0	/
Aopingzhen	3	62	95.38	2	28	93.33	1	5	83.33	0	27	100	0	2	100
Tongjizhen	7	112	94.12	5	45	90	1	7	87.5	1	67	98.53	0	0	/
Zhihezhen	8	109	93.16	4	25	86.21	1	5	83.33	3	72	96	0	1	100
Xiaoyudongzhen	4	50	92.59	3	50	94.34	1	0	0	0	0	/	0	0	/
Shengpingzhen	10	105	91.3	7	15	68.18	0	0	/	3	90	96.77	0	0	/
Hongyanzhen	2	20	90.91	2	6	75	0	2	100	0	12	100	0	0	/
Guihuazhen	1	7	87.5	1	4	80	0	2	100	0	0	/	0	1	100
Gexianshanzhen	7	40	85.11	4	11	73.33	3	5	62.5	0	24	100	0	0	/
Longfengzhen	7	32	82.05	3	27	90	4	5	55.56	0	0	/	0	0	/
Tianpengzhen	93	332	78.12	15	92	85.98	3	12	80	75	228	75.25	0	0	/
Mengyangzhen	12	37	75.51	6	14	70	6	1	14.29	0	21	100	0	1	100
Lichunzhen	54	109	66.87	54	108	66.67	0	1	100	0	0	/	0	0	/
Bailuzhen	1	0	0	0	0	/	1	0	0	0	0	/	0	0	/

## Data Availability

Not applicable.

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
