# Peer review of "Molecular Transmission Network of Newly Reported HIV Infections in Pengzhou, Sichuan Province: A Study Based on Genomics and Spatial Epidemiology"

_ijerph, 2023, doi:10.3390/ijerph20032523_

Round 1

Reviewer 1 Report

This is a well conducted research, well presented. The English language can still be improved, and the content can be presented in a more dense format. The abstract for instance is too extensive and could be shortened. Also in the discussion part we read sentences that are really difficult to understand: too complicated, sometimes too much repetition. The text would greatly benefit from editing by an English native speaker.

Author Response

Response: Thank you much for your suggestion. Due to limited budget, we did not invite English native speaker who are familiar with the study field to edit our manuscript. We try our best to modify the manuscript, especially the discussion part.

Reviewer 2 Report

Dear authors, 

Thank you for your interesting manuscript. I have the following suggestions for improvement:

- Please elaborate the percentage of recruit/sample size in comparison to all eligible newly detected PLWH. 

- Please mention how you make sure the same size is rather representative (m/f,  risk factors, age among other variables). 

- IIf the above may have potentially impact on biais, please add them as the limitations of the study

- Please elaborate the data confidentiality and how the participants would have or have not been informed of the results of the study.

Thanks and regards

Author Response

Reviewer 2

Thank you for your interesting manuscript. I have the following suggestions for improvement:

1.- Please elaborate the percentage of recruit/sample size in comparison to all eligible newly detected PLWH.

Response:From March 2019 to August 2021, there were a total of 720 newly reported HIV infections , of which 484 were recruited, accounting for 67.22 %.

2.- Please mention how you make sure the same size is rather representative (m/f,  risk factors, age among other variables).

Response:We recruited 67.22 % of the total newly reported HIV infections who have similar characteristics such as gender, age, occupation, culture and marital status with literatures. More details are the followings:

(1):The proportion of new HIV reports among middle-aged and elderly people aged 50 and above in China is increasing year by year, and China is very concerned about HIV infection among middle-aged and elderly people aged 50 and above[1]. It has been reported that[2], the number of people aged 50 and over is increasing, because young people with AIDS have longer survival and older age, and more and more elderly people are infected with AIDS. As of December 2020, there are about 8.1 million cases of middle-aged and elderly people over 50 years old[3].It is expected that by 2030, the proportion of some countries will rise to 73 %[4]. From 2015 to 2019, the number of newly reported PLHIV infections in middle-aged and elderly people over 50 years old in China increased from 32,850 to 66010[5].At the same time, 11056 newly reported HIV / AIDS cases in the age group of 50 years and above in Sichuan Province in 2021 accounted for 60.90 % of the newly reported cases in that year, and the proportion of infected people in this study area reached 70 %. The age group of 50 and above is the group that needs to be focused on, and it is also the main group of research. Therefore, this time, 50 yr is used as the dividing point to study the differences between the middle-aged and elderly groups and young adults.

   (2)The basis for the diagnosis of AIDS in adults and adolescents over 15 years old includes HIV infection and CD4 + T lymphocyte count < 200 / μl[6]. When CD4 count < 200 / μl, it is the final stage after HIV infection. The main clinical manifestations are HIV-related symptoms, signs and various opportunistic infections and tumors. Therefore, the choice of CD4 count before treatment to 200 / μl as the division limit[7], which should be representative.

(3) Select the number of cluster nodes in the molecular network ≥ 2 as the dependent variable. First, because the quality of AIDS prevention and treatment in Chengdu is higher than that in other regions, the quality of treatment is also higher than that in other regions. Therefore, it is considered that the infection with 2 or more cases of transmission is a high-risk communicator[8].

3- If the above may have potentially impact on biais, please add them as the limitations of the study

Response:The above division of variable sizes lacks sufficient research and practice in Pengzhou, which may cause a certain degree of bias in the results. It is proposed to explain the limitations of the study.

4- Please elaborate the data confidentiality and how the participants would have or have not been informed of the results of the study.

Response:Reference strains for this study are from the Los Alamos HIV sequence database [ https://www.hiv.lanl.gov/]. Other datasets used and / or analyzed in the current study are available upon reasonable request from the respective authors.

All participants sign a written informed consent prior to the sample donation requesting strict adherence to the principle of data confidentiality, together with reasonable access to research results, and all researchers sign data confidentiality agreements

Reviewer 3 Report

HIV-1 molecular epidemiological surveys of recent infections are important for understanding the real-time dynamics of the HIV-1 epidemic, and this is an interesting manuscript utilizing this methodology.

ABSTRACT: There is no need to include acronyms if you will not use it again. For instance, you defined PCR and ML, but didn’t use it again. PLWH refers to ‘People living with HIV’, not to ‘newly reported HIV infections’ (there is no acronym here).

INTRODUCTION

Please review your estimates. According to the latest UNAIDS Report, 84.2 million [64.0 million–113.0 million] people have become infected with HIV since the start of the epidemic - not 76 million as you referred. You should start your introduction describing the world HIV/AIDS scenario, then the epidemiological aspects of it in China and then present the specificities of the Sichuan Province. This will give readers a better understanding of your local epidemiological scenario. It is December 2022, all your epidemiological data should be updated accordingly - you are currently presenting data from early 2019.

You need to improve your introduction, please include more information about: (1) the HIV/AIDS epidemic, with current data; (2) the importance of utilizing molecular biology to better understand epidemic dynamics; (3) include findings from molecular biology studies conducted in the world and in China - there are several manuscripts available. See, for instance: https://pubmed.ncbi.nlm.nih.gov/36032035/; https://pubmed.ncbi.nlm.nih.gov/27080587/; https://www.ncbi.nlm.nih.gov/pmc/articles/PMC8946679/; among many others.

Overall the introduction is not adequate and should be thoroughly revised. Those brief two paragraphs do not present adequately and thorough your research problem and what we already know about it.

METHODS

Please include a brief paragraph describing major socio-demographic characteristics of the Sichuan Province - many IJERPH readers might not be familiar with this province.

Statistical Analysis: Please include what statistical tests you used.

RESULTS

I think your Table 1 should be the first thing to be presented. It is important to describe the demographics and epidemiological characteristics of your study population first, and then present more specific molecular biology data.

One doubt: Why the authors used 50yr as the cutoff? This seem to put together a highly diverse population of adolescents, young adults and middle aged patients… Please explain.

Each figure should have a title, please revise. All figures should also be cited in-text, please revise accordingly.

Figure 2 is too small and hard to read.

Spacial analysis: Here I think you should include only the maps, both tables should be added as supplementary files. IJERPH has a large international audience that might not be familiar with the townships of Pengzhou city, therefore a map makes it easier to see identified hot spots and connections between areas.

DISCUSSION

Could you please compare your findings with other studies conducted in China? Are the same HIV subtypes more prevalent in Mainland China? Is there any specific characteristic of the HIV/AIDS local epidemic that is singular to the Sichuan Province?

Findings from the spatial analysis could be compared to availability of health facilities providing HIV preventive services (e.g. condom distribution, PrEP, PEP) and HIV testing? And finally, what are the study implications for future public health strategies in the Sichuan Province?

REFERENCES

Please avoid self citation

Thanks for the opportunity to review your manuscript.

Author Response

Reviewer 3

1.HIV-1 molecular epidemiological surveys of recent infections are important for understanding the real-time dynamics of the HIV-1 epidemic, and this is an interesting manuscript utilizing this methodology.

2.ABSTRACT: There is no need to include acronyms if you will not use it again. For instance, you defined PCR and ML, but didn’t use it again. PLWH refers to ‘People living with HIV’, not to ‘newly reported HIV infections’ (there is no acronym here).

Response:Modified

3.INTRODUCTION

Please review your estimates. According to the latest UNAIDS Report, 84.2 million [64.0 million–113.0 million] people have become infected with HIV since the start of the epidemic - not 76 million as you referred. You should start your introduction describing the world HIV/AIDS scenario, then the epidemiological aspects of it in China and then present the specificities of the Sichuan Province. This will give readers a better understanding of your local epidemiological scenario. It is December 2022, all your epidemiological data should be updated accordingly - you are currently presenting data from early 2019.

You need to improve your introduction, please include more information about: (1) the HIV/AIDS epidemic, with current data; (2) the importance of utilizing molecular biology to better understand epidemic dynamics; (3) include findings from molecular biology studies conducted in the world and in China - there are several manuscripts available. See, for instance: https://pubmed.ncbi.nlm.nih.gov/36032035/; https://pubmed.ncbi.nlm.nih.gov/27080587/; https://www.ncbi.nlm.nih.gov/pmc/articles/PMC8946679/; among many others.https://pubmed.ncbi.nlm.nih.gov/36032035/;https://pubmed.ncbi.nlm.nih.gov/27080587/;https://www.ncbi.nlm.nih.gov/pmc/articles/PMC8946679/;等等。

Overall the introduction is not adequate and should be thoroughly revised. Those brief two paragraphs do not present adequately and thorough your research problem and what we already know about it.

Response:Thank you very much suggetsion. We have revised the introduction part as suggested.

  1. METHODS

Please include a brief paragraph describing major socio-demographic characteristics of the Sichuan Province - many IJERPH readers might not be familiar with this province.

 Statistical Analysis: Please include what statistical tests you used.

Response:Thank you very much suggetsion. We have revised the manucript as suggested. 

  1. RESULTS

5.1 I think your Table 1 should be the first thing to be presented. It is important to describe the demographics and epidemiological characteristics of your study population first, and then present more specific molecular biology data.

Response:table 1 combines the demographics and epidemiological characteristics of your study population(column “total” and “n%”) and “Analysis of influencing factors of Pengzhou molecular network cluster nodes ≥ 2 together”.

5.2 One doubt: Why the authors used 50yr as the cutoff? This seem to put together a highly diverse population of adolescents, young adults and middle aged patients… Please explain.

Response:The proportion of new HIV reports among middle-aged and elderly people aged 50 and above in China is increasing year by year, and China is very concerned about HIV infection among middle-aged and elderly people aged 50 and above[1]. It has been reported that[2], the number of people aged 50 and over is increasing, because young people with AIDS have longer survival and older age, and more and more elderly people are infected with AIDS. As of December 2020, there are about 8.1 million cases of middle-aged and elderly people over 50 years old[3].It is expected that by 2030, the proportion of some countries will rise to 73 %[4]. From 2015 to 2019, the number of newly reported PLHIV infections in middle-aged and elderly people over 50 years old in China increased from 32,850 to 66010[5].At the same time, 11056 newly reported HIV / AIDS cases in the age group of 50 years and above in Sichuan Province in 2021 accounted for 60.90 % of the newly reported cases in that year, and the proportion of infected people in this study area reached 70 %. The age group of 50 and above is the group that needs to be focused on, and it is also the main group of research. Therefore, this time, 50 yr is used as the dividing point to study the differences between the middle-aged and elderly groups and young adults.

6.Each figure should have a title, please revise. All figures should also be cited in-text, please revise accordingly.

Response:The corresponding modifications have been made on the graph name

7..Figure 2 is too small and hard to read.

Response: Modified

  1. Spacial analysis: Here I think you should include only the maps, both tables should be added as supplementary files. IJERPH has a large international audience that might not be familiar with the townships of Pengzhou city, therefore a map makes it easier to see identified hot spots and connections between areas.

Response:At the time of uploading, Schedule 1 and 2 are used as supplements. The spatial transmission map analyzes several subtypes of strains. If a map is used, it will appear to be very close and cannot distinguish the spatial transmission of each subtype. Therefore, a in Figure 3 shows the township area and clustering of Pengzhou, b shows the spatial transmission of all subtypes, and c-f shows the spatial transmission of CRF01 _ AE, B, CRF07 _ BC and CRF08 _ BC, respectively.

9.DISCUSSION

 Could you please compare your findings with other studies conducted in China? Are the same HIV subtypes more prevalent in Mainland China? Is there any specific characteristic of the HIV/AIDS local epidemic that is singular to the Sichuan Province?

Response:Thank you very much suggetsion. We have revised the discussion part as suggested.

The results of previous global studies showed that during 2010-2015, the prevalence of strains was mainly subtype C ( 46.6 % ), followed by subtype B ( 12.1 % ), subtype A ( 10.3 % ), CRF02 _ AG ( 7.7 % ), CRF01 _ AE ( 5.3 % ), subtype G ( 4.6 % ) and subtype D ( 2.7 % ). From 2010 to 2015, the epidemic strains in Western Europe, North America and Central Europe began to be dominated by subtype B, and the proportion gradually decreased in the later period. The epidemic strains in Eastern Europe and Central Asia were dominated by subtype A ( 52.8 % ), and the epidemic strains in Southeast Asia were CRF01 _ AE. The prevalence rate was the highest ( 72.8 % )[22]. The results of the latest fourth molecular epidemiological survey in China showed that CRF07 _ BC ( 41.3 % ) was the main epidemic strain in China, followed by CRF01 _ AE ( 32.7 % ), CRF08 _ BC ( 11.3 % ) and subtype B ( 4.0 % )[23].In 2014, the molecular epidemiology of Sichuan Province was also dominated by CRF07BC ( 58.22 % ) and CRF01AE ( 29.50 % ) subtypes[24].Among them, CRF07BC and CRF01AE had the highest rate of molecular network entry[25, 26] . The epidemic strains in this study were mainly CRF01AE ( 51.19 % ) and CRF07BC ( 34.34 % ), but the molecular network entry rate was the highest in subtype B, showing aggregation infection, which was the unique epidemic feature of this study.

10.Findings from the spatial analysis could be compared to availability of health facilities providing HIV preventive services (e.g. condom distribution, PrEP, PEP) and HIV testing? And finally, what are the study implications for future public health strategies in the Sichuan Province?

Response:By combining spatial analysis, it can help to understand which regions are more closely related to each other and have higher transmission risks, and provide scientific basis for inter-regional joint prevention and control. The results of visual analysis of medical resources show that the number of HIV testing laboratories in the local townships and streets is mainly concentrated in the central town Tianpeng Town and its surrounding towns. The number of HIV molecular clusters and the intensity of HIV regional transmission in the south are also high. The network gathering place and the medical facility construction configuration show the same aggregation trend, but it cannot reflect the effect of AIDS prevention and intervention. It is necessary to increase the corresponding intervention facilities and intervention efforts, such as the establishment of joint prevention and control between townships and towns, communication mechanism and so on. However, the failure to collect such data is also the limitation of this study.

11.REFERENCES

Please avoid self citation

Response: Modified
